# Pretreatment Adherence to a Priori-Defined Dietary Patterns Is Associated with Decreased Nutrition Impact Symptom Burden in Head and Neck Cancer Survivors

**DOI:** 10.3390/nu13093149

**Published:** 2021-09-09

**Authors:** Christian A. Maino Vieytes, Alison M. Mondul, Sylvia L. Crowder, Katie R. Zarins, Caitlyn G. Edwards, Erin C. Davis, Gregory T. Wolf, Laura S. Rozek, Anna E. Arthur

**Affiliations:** 1Division of Nutritional Sciences, University of Illinois at Urbana-Champaign, Urbana, IL 61801, USA; cam17@illinois.edu (C.A.M.V.); cke5143@psu.edu (C.G.E.); erin_davis1@urmc.rochester.edu (E.C.D.); 2Department of Epidemiology, University of Michigan, Ann Arbor, MI 48109, USA; amondul@umich.edu; 3Department of Health Outcomes and Behavior, Moffitt Cancer Center, Tampa, FL 33612, USA; sylvia.crowder@moffitt.org; 4Department of Food Science and Human Nutrition, University of Illinois at Urbana-Champaign, Urbana, IL 61801, USA; 5Department of Environmental Health Sciences, University of Michigan, Ann Arbor, MI 48109, USA; kmrents@umich.edu (K.R.Z.); rozekl@umich.edu (L.S.R.); 6Department of Nutritional Sciences, The Pennsylvania State University, University Park, PA 16802, USA; 7Department of Pediatrics, Division of Allergy and Immunology, School of Medicine and Dentistry, University of Rochester, Rochester, NY 14642, USA; 8Department of Otolaryngology, University of Michigan, Ann Arbor, MI 48109, USA; Gregwolf@med.umich.edu; 9Department of Dietetics and Nutrition, University of Kansas Medical Center, Kansas City, KS 66160, USA

**Keywords:** survivorship, cancer, nutritional epidemiology, nutrition impact symptoms

## Abstract

Dietary intake is understood to contribute to nutrition impact symptoms (NIS) in patients with head and neck squamous cell carcinoma (HNSCC). The purpose of this study was to evaluate the performance of four a priori-defined diet quality indices on the presence of NIS 1 year following diagnosis using data on 323 participants from the University of Michigan Head and Neck Specialized Program of Research Excellence (UM-SPORE). Pretreatment dietary intake was measured before treatment initiation using a food frequency questionnaire. NIS were measured along seven subdomains. Multivariable binary logistic regression models were constructed to evaluate relationships between pretreatment scores on a priori-defined diet quality indices (AHEI-2010, aMED, DASH, and a low-carbohydrate score) and the presence of individual symptoms in addition to a composite “symptom summary score” 1-year postdiagnosis. There were several significant associations between different indices and individual NIS. For the symptom summary score, there were significant inverse associations observed for aMED (OR_Q5-Q1_: 0.36, 95% CI: 0.14–0.88, *p*_trend_ = 0.04) and DASH (OR_Q5-Q1_: 0.38, 95% CI: 0.15–0.91, *p*_trend_ = 0.02) and the presence of NIS 1-year postdiagnosis. Higher adherence to the aMED and DASH diet quality indices before treatment may reduce NIS burden at 1-year postdiagnosis.

## 1. Introduction

Head and neck squamous cell carcinoma (HNSCC) accounts for roughly 4% of all new cancer diagnoses in the United States [1]. HNSCC is commonly diagnosed in the oral cavity, oropharynx, hypopharynx, and larynx and is associated with lifetime exposure to tobacco and alcohol consumption and infection with particular strains of human papillomavirus (HPV) implicated, generally, in tumors affecting the oropharynx [2,3]. In addition to symptomatology arising from tumor morphology and location, side effects that impact food and oral intake due to cancer treatment are also highly prevalent in this population. Nutrition impact symptoms (NIS), as they are termed, include but are not limited to dysgeusia, ageusia, xerostomia, pain, dysphagia, dental problems, mucositis, and trismus [4]. An estimated 90% of HNSCC patients develop acute NIS due to their cancer treatment [5]. Consequently, this symptom burden may perpetuate significant physical, emotional, and psychological issues, hampering the overall quality of life (QOL) and QOL around eating [6,7]. Nonetheless, evidence on chronic NIS following treatment remains scant. The report by Ganzer et al. found that NIS, including dysphagia, xerostomia, and altered taste, persisted after three years postchemotherapy in a mixed-methods study of 10 long-term HNSCC survivors [8]. Chronic NIS may pose significant nutritional consequences, including reduced nutrient intake and impaired nutritional status, which is noteworthy, given that this population is disproportionately affected by high rates of cancer cachexia [9,10].

A posited etiologic mechanism for NIS involves inflammation secondary to cancer treatment and, in particular, treatment with targeted radiation [11]. While previous studies have provided that consuming a balanced and healthful diet rich in fruits and vegetables before treatment may abate inflammation and chronic NIS, research has yet to identify a generalizable dietary pattern that confers protection by mitigating inflammation after diagnosis and throughout cancer treatment [9]. Using a multivariate approach for characterizing a posteriori dietary patterns, our research team reported associations between consuming a primarily “prudent” diet, rich in fruits and vegetables, at pretreatment and reduced symptom burden at 1-year postdiagnosis [9]. Nevertheless, the purpose of the present analysis was to assess the capacity of a priori-defined diet quality indices to predict symptom burden 1-year postdiagnosis, which, for many participants, comes after the initiation and completion of treatment protocols.

A priori diet quality indices are generally used to measure adherence to a set of dietary recommendations or guidelines, in contrast to a posteriori methods implemented to characterize eating behaviors from sample dietary data [12]. An advantage that a priori indices have over their a posteriori counterparts is their generalizability and their facility for policy adaptation. Moreover, index scores calculated on different study samples are directly comparable, whereas a posteriori patterns are not, since they characterize a given sample. In this analysis, we chose to examine the performance of four a priori diet quality indices reported previously in the scientific literature. The results of this analysis could be harnessed to tailor dietary recommendations for HNSCC patients to abrogate the incidence of NIS and would form, to our knowledge, the first study of a priori diet quality indices and their relationship to NIS in HNSCC following diagnosis and treatment. The study hypotheses were that higher adherence to each of the a priori diet quality indices examined corresponds to lower self-reported chronic NIS 1-year postdiagnosis.

## 2. Materials and Methods

### 2.1. Study Population

This was a secondary analysis of clinical and dietary data gathered on participants in the University of Michigan Head and Neck Specialized Program of Research Excellence (UM-SPORE) cohort. UM-SPORE is a prospective, longitudinal cohort study of newly diagnosed patients with HNSCC who presented with primary malignancies in the oral cavity, oropharynx, hypopharynx, or larynx and entered the study before the initiation of any treatment for primary HNSCC. Recruitment was conducted through the UM Hospital System and took place between November 2008 through October 2014, whereby newly diagnosed HNSCC cases were screened and solicited consent for inclusion into the study. Informed consent was obtained from all subjects involved in the study. Exclusion criteria for the study are detailed as: (i) age less than 18 years; (ii) being pregnant; (iii) being a non-English speaker; (iv) having a previously diagnosed mental disorder; (v) previous or concomitant diagnosis of a tumor in the non-upper aerodigestive tract; and (vi) previous diagnosis with another form of primary HNSCC within the last five years. Upon entry, participants completed a baseline (pretreatment) food frequency questionnaire (FFQ) and survey questionnaires ascertaining lifestyle and epidemiologic characteristics. Survey measures included history of other identified comorbid conditions, smoking status, drinking status, sleep, physical activity, and depression. All baseline (pretreatment) data collection was conducted before the initiation of any treatment protocol for HNSCC, and participants were subsequently followed longitudinally. Annual reviews of electronic medical records were used to extract clinical factors, including cancer stage, site, and treatment protocol data.

There were 380 participants with baseline and 1-year NIS data, which also had complete baseline/pretreatment FFQ data. Further exclusions included those with missing body mass index (BMI) data (*n* = 8) and those missing data on any other covariates used in the study (*n* = 3). Subjects reporting caloric intakes of >5000 kcal/d or <500 kcal/d (*n* = 5) were excluded on the premise that these levels of intake are likely implausible, making these observations unreliable, which may bias the final results [13]. Furthermore, participants with tumors at sites other than the larynx, oropharynx, hypopharynx, or oral cavity (*n* = 31), missing full pages of their pretreatment FFQ (*n* = 9), and having greater than 70 blank responses on their FFQ (*n* = 1) were excluded from the analysis [14]. The final analytic sample comprised 323 participants. All study procedures were executed in compliance with standards approved by the University of Michigan Institutional Review Board (IRB approval number, for which consent was granted for obtaining and analyzing the data, is HUM00042189) and complied with the Helsinki Declaration of 1975.

### 2.2. Predictors: Pretreatment a Priori Diet Quality Index Scores

Baseline dietary intake data were collected using the self-administered 2007 Harvard Adult FFQ, a 131-item semiquantitative FFQ formulated to assess the usual intake of select foods, beverages, and supplements and is used to compute a profile of average nutrient intake for a given participant [15,16]. This method affords a practical approach for ranking the participant sample based on relative food and nutrient intake. Participants were asked to complete the questionnaire based on what they believe their usual intakes for select foods and beverages were over the past year. This was prompted through inquiries that accounted for standard portion sizes and frequency (e.g., 2–4 times per week, 1 medium banana). Nutrient intakes were computed by taking proportional weights corresponding to the frequency of intake selected for a given food item, multiplying by the nutrient value for the portion/serving size established on the questionnaire, and then summing across all foods [15]. Nutrient composition values were estimated using the Harvard nutrient database.

Four frequently cited a priori-defined diet quality indices were chosen for the analysis. These included the Dietary Approaches to Stop Hypertension (DASH), the Alternate Mediterranean Diet Index (aMED), the Alternative Healthy Eating Index-2010 (AHEI-2010), and a low-carbohydrate diet index. The choice to use these particular indices arose from their widespread use in the nutritional epidemiology literature and, specifically, within the context of chronic disease risk and management [17,18,19]. Nutrient- and item-specific intake levels were estimated from the administered Harvard FFQ data and used to calculate diet quality index scores.

The DASH diet has previously been described and is extensively documented as a treatment protocol in hypertension. This dietary pattern emphasizes fruits, vegetables, whole grains, low-fat dairy, nuts, and legumes while limiting intakes of red meat, sweets, and sugar-sweetened beverages [20]. Concerning nutrient intakes, the DASH diet is characterized by reduced intakes of salt (sodium chloride), saturated and total fat, and increased intake of foods with high mineral (primarily potassium and magnesium) and micronutrient value. Calculation of the DASH diet scores was adapted to this cohort using the framework described by Fung et al. [21]. The operationalization of this dietary protocol ranks participants according to their average intake in 8 select food group components: fruits, vegetables, nuts/legumes, low-fat dairy products, whole grains, sodium, red and processed meats, and sugar-sweetened beverages. Scores for each of the first five listed components were taken as the quintile ranking for a participant for that food group. Component scores for the latter three components were assigned antagonistically. Individuals scoring within the highest quintile of intake were given a score of “1”, whereas those residents to the lowest quintile of intake were given a score of “5”. Summing scores across all components allowed us to arrive at the final composite score, which had a maximum value of 40.

The aMED diet quality index is based on the operationalization provided by Fung et al. [18]. The traditional Mediterranean diet pattern has been characterized by high intakes of fruits, vegetables, breads, cereals, legumes, high-quality fats (primarily olive oil) [22]. Moderate to low intakes of red meat, fish, low-fat dairy, and alcohol (primarily wine) also make up intrinsic components. This dietary pattern is further stipulated by its limiting of foods with a processed origin. Calculation of the aMED score considers intake levels of 9 components that were obtained from participant FFQ data: vegetables, legumes, fruits, nuts, whole grains, red or processed meats, fish, alcohol, and the ratio of monounsaturated/saturated (UFA/SFA) fat intake. Component scores were based on a participant’s rank relative to the median intake for that component. That is, those with intakes greater than the median were given a score of “1”, while those falling below the median were given a score of “0”. For the meat component, falling above the median intake resulted in a score of “0”, while ranking below the median gave participants a score of “1”. Alcohol intakes between 5 and 15 g/d were designated a score of “1” for the alcohol component. The final composite score was computed by summing scores across all of the 9 components, with a maximum score of 9.

The AHEI was developed in 2002 as an alternative to the Healthy Eating Index (HEI), which operationalized the 1995 iteration of the Dietary Guidelines for Americans, and was tailored with the intention of being a more robust indicator of chronic disease risk [19]. This diet quality index was subsequently updated in 2010 and emphasized similar food components to the aforementioned indices with additional foci on trans fat (as a percentage of total energy intake), polyunsaturated fatty acids (as a percentage of total energy intake), and n − 3 (EPA + DHA) fatty acid intake. Similar to the calculation of aMED, it awards points for the moderate consumption of alcohol. Operationalizing the index relies on mapping intakes for each food category to a scale ranging from 0 to 10. The scoring algorithm has been previously described by Chiuve et al., and the maximum attainable score for any given participant is 110 [19].

Finally, a low-carbohydrate index, standing in as a proxy for a ketogenic diet, was computed as previously described by Halton et al. [23]. Briefly, percentages of energy intake from each of carbohydrate, fat, and protein were calculated for the study subjects, and they were subsequently partitioned and ranked according to quantiles of intake for each category. For the protein and fat categories, scores were allocated congruently with participant rank (i.e., a rank of “1” was commensurate to a score of “1”). For the carbohydrate score, scores were allocated antagonistically (i.e., a rank of 10 resulted in a score of “0”). A theoretical maximum score of 30 was attainable for this index.

### 2.3. Covariates

Sociodemographic covariates included age (modeled continuously), sex (modeled dichotomously), and education status (coded as less than or equal to high school or some college or more). Behavioral characteristics included in our models consisted of smoking status (modeled categorically as never, former, or current). The clinical variables were BMI (modeled dichotomously as <25—normal or underweight—or ≥25—overweight or obese), tumor HPV infection status (modeled as positive, negative, or equivocal/missing test), cancer stage (modeled dichotomously as 0, I, II or III, IV), and tumor site (modeled categorically as larynx, oral cavity, oropharynx, or hypopharynx). All models examining 1-year NIS variables as their outcome were adjusted for their baseline categorical groupings, derived from their corresponding scale values at baseline (groups were dichotomized as outlined below). Lastly, all models adjusted for total energy intake by including total calories (kcal) as a continuous variable. Treatment modality, sex, and drinking status were given a priori consideration for inclusion but were omitted given that they previously were shown to be highly correlated with other covariates among this patient cohort [24]. Nonetheless, subanalyses included examining associations amongst the different treatment levels to account for any varying effects of particular treatment protocols on NIS, as detailed below.

### 2.4. Outcomes: NIS at 1-Year Postdiagnosis

Six levels of NIS (trismus, xerostomia, dysphagia with liquids, dysphagia with solid foods, difficulty chewing, and taste perception) were assessed and quantified using the UM Head and Neck Quality of Life (QOL) Questionnaire developed and validated for use in this patient population by Terrell et al. [25]. This 37-item survey evaluates the landscape of HNSCC patient QOL by emphasizing four meaningful domains: communication, eating, emotion, and pain. Six items encompassing the eating subdomain were used to measure the aforementioned outcome variables, and available responses were provided on a discrete 5-point scale from “not at all bothered” (given a numerical score of “1”) to “extremely bothered” (indicating a score of “5”). An additional item, assessing the burning pain and discomfort that characterizes mucositis, was included and extracted from the pain domain of the questionnaire. Mucositis has been shown to impact dietary intake and thus was included in the analysis for this reason [9,26]. Responses to each of these individual items were dichotomized into categories “not at all bothered” and “slightly to extremely bothered”, as was previously done by members of our research team [27]. A composite symptom summary measure using 1-year NIS data was developed by taking the sum of participant responses across these seven items. A maximum value of 35 was attainable on this index, representing the most severe symptom burden. Subsequently, participants were dichotomized into groups based on a median-split, with NIS symptom summary score <12 defined as a low-symptom burden and scores ≥ 12 defining those experiencing a high-symptom burden. This threshold value was chosen and based on a previous operationalization of this scale [27].

### 2.5. Statistical Analysis

Descriptive analyses were performed, examining frequencies and means across demographic, behavioral, and clinical factors. The mean 1-year NIS summary score was evaluated across relevant epidemiologic characteristics and tabulated. One-way analysis of variance (ANOVA) was used to assess for significant differences in mean 1-year NIS summary scores across levels of the characteristics examined. Tukey’s post-hoc mean separation test was implemented to partition significantly different groups within the different characteristics examined. Bivariate relationships were evaluated with Pearson correlation coefficients computed amongst the four a priori diet quality indices chosen for the analysis.

Continuous and discrete scores from the four a priori diet quality indices were categorized by quintiles. Multivariable binary logistic regression models were fit to evaluate the associations between each a priori diet quality index score and the relevant outcomes. In total, there were eight models constructed for each diet quality index score: (i) a separate model for each of the seven symptoms introduced above and (ii) a model examining the dichotomized 1-year NIS summary score rank as the outcome of interest. All primary analytical models adjusted for participant age, smoking status, BMI category, total calories, educational status, HPV status, cancer stage, tumor site, and the corresponding symptom score group at baseline. All analyses used the lowest quintile (Q1) of intake as the referent group. Odds ratios (OR) and their 95% confidence intervals were computed and tabulated. Tests for linear trend were assessed by assigning the median value of a participant’s corresponding quintile and modeling that term as a continuous variable. To assess whether single food group or nutrient categories (measured as either servings per day or total mass in grams) would be able to recapitulate results from models using the composite indices, sixteen additional models were fit, using quintiles of intake for different categories of foods and nutrients that, together, make up the components of the calculated indices.

Stratified analyses (for the outcome of 1-year NIS symptom summary score) were conducted and examined the tested associations across strata for baseline BMI, smoking status, cancer stage, education, tumor site, and treatment modality (radiation versus no radiation used). Stratified models used a truncated set of covariates (age, smoking status, stage, total calories, and HPV status) to ensure adequate model fit and preserve statistical power with the smaller subsets. A sensitivity analysis, used to evaluate for the potential of reverse causality explaining our results, was conducted, where models were fitted separately on subjects reporting no symptoms at study entry/pretreatment (*n* = 72) and those with at least some degree of symptomatology at study entry (*n* = 251). Furthermore, restricted cubic spline models were fit to visually ascertain the observed relationships, examine linearity, and assess dose-dependence between each dietary predictor and the odds of 1-year NIS summary score ≥12. These models used four interior knots, set at the scores corresponding to quintiles of the respective diet quality index. The median of the lowest quintile of intake for each diet quality index was used as the referent value when computing odds ratios from these models. All analyses were conducted at *α* = 0.05 and performed in RStudio version 1.4.

## 3. Results

### 3.1. Sample Characteristics

Table 1 provides descriptive statistics and means for the analytic cohort. The average age of the analyzed sample was 60.4 years. Generally, age was larger in the highest quintile of intake relative to the lowest for each of the four diet quality indices examined. This cohort contained a majority of males (*n* = 254; 78.6%). There tended to be a higher proportion of females within the highest quintile of the diet quality indices compared to the lowest quintile. Of note, most participants identified as non-Hispanic white (*n* = 310; 96.9%). BMI was variable across the different indices, and it had no clear relationship with quintiles of the diet quality indices. Regarding the behavioral variables, proportions of current smokers tended to be higher within the lowest quintile of the examined indices and were most pronounced for AHEI-2010 and DASH, whereas former and never smoker proportions were higher in the highest quintiles of the aMED, AHEI-2010, and DASH indices. Drinking status across quintiles of the indices also suggested higher proportions of current consumption within the lowest quintile compared to the highest (this was true for all indices but most pronounced for aMED and DASH). The differences seen in HPV status across quintiles of those indices appeared to follow the patterns in smoking status, to an extent except for the low-carbohydrate index. There were no other appreciable differences in distributions of participant characteristics.

### 3.2. NIS Symptom Summary Scores and Potential Confounders

Concerning the 1-year NIS symptom summary score, the primary analytical outcome, individuals with lower educational background (≤high school completed) tended to have a significantly, albeit slightly, higher symptom burden score (*p* < 0.01). A significant association within tumor site was also identified (*p* < 0.01). More pronounced symptomatology was reported in participants with oropharynx vs larynx tumors. Subjects classified with stage III or IV tumors had significantly higher NIS symptom scores than those with tumors staged 0, I, or II (*p* < 0.01). Additionally, several significant differences in symptom scores were noted across treatment classes (*p* < 0.01). All of these described differences are documented in Table 2.

### 3.3. Diet Quality Scores

Summary statistics for each of the diet quality indices examined and results of the correlational analysis are found in Table 3 and Appendix A, respectively. Median scores for the AHEI-2010, aMED, DASH, and the low-carbohydrate index with the analyzed sample were 58.54 (Min: 22.85, Max: 89.14), 4 (Min: 0, Max: 9), 24 (Min: 10, Max: 37), and 15 (Min: 0, Max: 30), respectively, and were generally commensurate to the estimated sample means. aMED, DASH, and AHEI-2010 scores all shared Pearson correlation coefficients suggestive of moderately strong and positive relationships. aMED and DASH scores were weakly and inversely correlated with the low-carbohydrate score, while the AHEI-2010 shared a weak, positive correlation with the low-carbohydrate index.

### 3.4. NIS Symptom Burden 1-Year Postdiagnosis

We evaluated the associations between consumption along a priori diet quality scores, derived from FFQs, using multivariable binary logistic regression. These results are referenced from Table 4. When examining the associations between baseline aMED diet quality index scores and responses to the seven NIS symptom scales at 1-year postdiagnosis, it was found that higher consumption along this index was strongly and inversely associated with dysphagia of liquids and, to a lesser extent, with dysphagia of solids, difficulty chewing, xerostomia, and mucositis. A strong and significant inverse relationship with the NIS 1-year symptom summary score was also observed (OR_Q5-Q1_: 0.36, 95% CI: 0.14–0.88, *p*_trend_ = 0.04). Closer adherence to the DASH protocol was significantly inversely associated with all 1-year symptom scales besides trismus, mucositis, and dysphagia solids. However, the parameter estimates in all of these models for each of quintiles 2–5 were suggestive of a protective association that failed to meet the threshold for statistical significance. The strongest inverse relationship was that with xerostomia (OR_Q5-Q1_: 0.27, 95% CI: 0.08–0.85, *p*_trend_ = 0.04). Higher consumption along the DASH index was also potently and significantly inversely associated with the 1-year NIS summary score (OR_Q5-Q1_: 0.38, 95% CI: 0.15–0.91, *p*_trend_ = 0.02). Associations along the AHEI-2010 index were more modest, with the strongest inverse association seen in xerostomia (OR_Q5-Q1_: 0.42, 95% CI: 0.14–1.21, *p*_trend_ = 0.04). There was no significant linear trend observed between consumption along AHEI-2010 and the 1-year symptom summary score, though the parameter estimate for the highest quintile of intake suggested a nonsignificant inverse association with a blunted effect estimate relative to the aMED and DASH indices. No significant inverse associations were noted between higher consumption along the low-carbohydrate index and any of the eight outcomes examined. Notably, there was a positive association between higher pretreatment consumption of the low-carbohydrate index and the odds of experiencing dysphagia with liquids at the 1-year mark (OR_Q5-Q1_: 2.47, 95% CI: 1.06-5.91, *p*_trend_ = 0.19). Most of the parameter estimates for the highest quintile of intake in the models considering individual symptoms as outcomes and low-carbohydrate diet score as the explanatory variable were greater than 1 except for trismus, taste, and mucositis. Though failing to meet the threshold for statistical significance, the parameter estimates for each of the second, third, and fifth quintiles of the low-carbohydrate index were all suggestive of a positive association with the 1-year NIS symptom summary score. The results of the set of restricted cubic splines analyses, modeling each diet quality index as a continuous variable and subsequently mapping index scores to their respective odds ratios from spline estimates, are visualized in Figure 1.

### 3.5. Analyses Using Single Nutrient Explanatory Variables

Modeling nutrient categories or food groups in place of the diet quality indices resulted in an abundance of non-significant figures (Table 5). There were a few notable exceptions. Total fruit consumption was strongly and inversely associated with the 1-year NIS summary score. Moreover, total nut consumption saw a very similar magnitude of association compared to fruit intake as the explanatory variable. A weaker inverse association was found for a model considering total n-3 fatty acid intake.

### 3.6. Subgroup and Sensitivity Analyses

Stratified analyses revealed disparities in several reported associations between diet quality and 1-year NIS symptom summary score among levels of BMI, smoking status, cancer stage, HPV status, education attained, tumor site, and treatment modality. The results of this analysis are found in Appendix A. There were few marked distinctions in the magnitude of associations compared to the results from primary analytical models. Those reporting lower educational status had significantly stronger magnitudes of association relative to those in higher education levels for the AHEI-2010, aMED, and DASH indices. Remarkably, the effect sizes in all tumor site levels were commensurate with the estimates from the overall model except for the association of the DASH index with 1-year NIS in the larynx subgroup (note that hypopharynx was omitted from this part of the analysis due to small sample size, *n* = 4). When examining levels of stage, the directionality and magnitude of the associations observed from the primary models were similar to those seen in either of the binary categories of stage, and the same can be said for the levels of HPV status examined. In consideration of participant treatment protocol, when evaluating models on participants who either received some form of radiation or those who did not, effect sizes for all indices were similar, and no substantial differences were appreciated. Finally, when models examining the relationships of baseline dietary indices on NIS 1-year postdiagnosis were fit on a sample subset of individuals entering the study without any symptoms at baseline, it was found that the parameter estimates were generally consistent with those from the primary analytical models including all sample subjects (Appendix A). Again, the strongest inverse associations within this subset were found in the higher quantiles of the DASH and aMED pretreatment index indices. In the case of the DASH index, these findings suggest a 76% reduction in the odds of significant NIS symptoms 1-year postdiagnosis for those entering the study without significant symptomatology and adhering closest to the DASH index compared to those also entering the study with no NIS but with the poorest adherence to the DASH protocol at study entry. The results were similar in those presenting with NIS at study entry, and this analysis was performed to evaluate for the possibility of reverse causation, whereby NIS present at pretreatment may have affected dietary adherence to the indices and, thus, distorted the relationship between explanatory variables and the outcomes in this analysis.

## 4. Discussion

We evaluated associations between four a priori-defined diet quality indices and found that greater adherence to the aMED and DASH dietary protocols during the year before treatment were each associated with diminished risks of experiencing self-reported NIS 1-year postdiagnosis. When analyzing individual NIS separately, it was found that several NIS were inversely associated with closer adherence to these indices. Overall, the aMED and DASH indices exhibited the most robust sets and the most numerous inverse associations when examining individual and overall NIS. The highest adherence category in each of the aMED and DASH indices exhibited a 64% and 62% reduction, respectively, in the odds of experiencing significant NIS burden at 1-year postdiagnosis relative to the lowest quintile of intake. These associations were followed in magnitude and significance by, to an even lesser extent, the AHEI-2010 index. In contrast, adhering to a ketogenic style, low-carbohydrate pattern was not associated with mitigations in self-reported NIS at the 1-year time point either when examining the NIS summary score or when analyzing any individual NIS. It was apparent that there may potentially be a detrimental influence of this low-carbohydrate diet quality index on symptom burden within the HNSCC population. However, these findings would need to be replicated and investigated further in clinical settings before these conclusions are ascertained.

When stratifying results by relevant participant characteristics reported at baseline, we found those significant associations were more pronounced in subjects with higher self-reported attained education status. Nevertheless, it should be acknowledged that participants in the group reporting greater attained education status (some college or more) had higher mean scores on all indices examined (results not shown). It is conceivable to hypothesize that participants in the lower education level were more likely to benefit from closer adherence to these indices, given potentially lower overall adherence. Reported associations for subjects who received radiation as part of their treatment protocol did not appear to diverge appreciably from those who did not receive radiation in their treatment regimen. This observed phenomenon appears to be somewhat inconsistent with reports in the literature of radiation-induced dysphagia and impaired swallowing function [12,28]. Notably, no single food groups, nutrients, or nutrient categories, other than total fruit and nut intakes, demonstrated associations as or more potent as those reported for the aMED or DASH indices. This highlights the ability of a priori diet quality indices to act as multidimensional factors that capture synergisms between food components and consequently significant associations that would otherwise go undetected in single nutrient or food group analyses.

We previously reported the associations between two a posteriori-derived dietary patterns and the same outcomes studied herein [27]. To our knowledge, this is the first study examining the relationships between the four chosen a priori-defined diet quality indices and those outcomes in the head and neck cancer patient population. Whereas empirically derived a posteriori dietary patterns, computed through methods such as principal components analysis or reduced rank regression, are patterns that are specific to and describe the populations under scrutiny, a priori-defined composites are routinely based on sets of predefined or established guidelines, thus underscoring the practicality of these dietary patterns across different populations. Consequently, identifying a priori indices that are particularly applicable and beneficial, in a population-specific manner, facilitates public health messaging and subsequent adoption of those protocols for given populations. The shrewdness of a priori or a posteriori indices has been extolled for its ability to more accurately model the complexity of diet as an epidemiologic or clinical exposure in human studies [29]. There are correlations or interactions amongst nutrients or foods that either blunt or bolster certain associations. Indeed, we observed this phenomenon in our study by modeling each nutrient or food category individually and finding that nutrients or food groups alone did not yield the degree of associations that were, instead, capitulated by modeling each diet quality indices. This analysis identified the aMED index as the most robust indicator of baseline diet quality that predicted reduced symptomatology 1 year following diagnosis. This performance was followed by that of the DASH index with very similar results.

The DASH diet was developed and demonstrated in 1997 as a dietary intervention for curbing hypertension-related sequelae [30]. Since its inception, the diet has been validated in numerous capacities and has drawn noteworthy recognition, being a focal point of national guidelines, particularly intended for those with hypertension [31]. Nonetheless, the tenets of the diet are typically in line with what many consider to be a “healthy” diet not only for hypertension, but for overall well-being and longevity. A Mediterranean diet is among the most frequently cited dietary patterns in the literature on longevity and chronic illness [32]. Though the guidelines for adhering to this regimen remain, in some respects, arcane and ambiguous, the principal emphasis of this pattern lies in the consumption of foods primarily of plant-based origin, olive oil as the chief lipid source, with minimal amounts of animal-derived foods and products [33]. Several methods of quantifying the eating patterns of populations inhabiting the Mediterranean Sea regions have been reported in the literature, and the aMED index represents one iteration. In many respects, DASH and aMED are similar, especially regarding how scores of adherence were tallied in this analysis. Both indices positively reward the consumption of fruits, vegetables, nuts, and legumes and castigate the consumption of red and processed meats. However, there are some notable differences. The DASH index includes additional components for low-fat dairy and sodium consumptions while not including components for fish and alcohol intake and a score for the ratio of fat sources in the diet like the aMED index stresses. In our study, food group subanalyses were not strongly associated with the outcome of NIS burden 1-year postdiagnosis. There was a nonsignificant downward trend in NIS burden across quintiles of low dairy intake. However, other food groups that differ across the indices, such as alcohol intake and sodium, were not predictive of the study’s primary analytical outcome. Again, this lack of association may relate to the relative advantage of using dietary patterns over single food groups. However, we posit that the difference in index algorithms is likely at play when the disparities between the performance of each of these indices are appreciated.

Alcohol intake is a known independent risk factor for incident HNSCCs, a family of cancers with strong ties to environmental etiologies, particularly in those cases lacking HPV seropositivity [34,35]. There have been few findings reported in how alcohol consumption impacts NIS for HNSCC cases with continuing alcohol use. Nevertheless, considering the findings that the aMED performed best amongst all indices examined and had an alcohol component appears to be somewhat consistent with those results reported by Potash et al. In their study of 283 HNSCC patients at 1-year postdiagnosis, it was reported that current “social drinkers” had the highest proportion of oral eating function compared to all other groups [36]. However, those labeled as “problem drinkers” were found to have compromised oral function compared to the social-drinking group. We can postulate variability in index performance is due to differences in the way alcohol consumption is rewarded. However, given the equivocal nature of the current evidence to back this conjecture, we propose that more research is warranted for delineating the effects of continued alcohol consumption, postdiagnosis, on HNSCC symptom burden. Further, it should also be noted that subject classification according to AHEI-2010 is based on absolute values of intake, whereas aMED and DASH are based on quantiles of intake within the analyzed sample, which reduces the risk of bias due to misclassification. The use of a method based on quantile classifications, rather than absolute values of intake, is substantiated by the fact that FFQ data, which was employed in this analysis, typically underperforms when the aim is to quantify absolute values of intake accurately but remains a viable method for ranking or distinguishing study subjects based on relative intake [14].

The potential beneficial effects imparted by higher adherence to either aMED or DASH indices are presumably mediated by the high consumption of plant-based foods, providing a food matrix that is ubiquitously filled with anti-inflammatory nutrients and phytochemical components. The consumption of these foods at baseline is plausibly linked to blunted symptomatology 1-year postdiagnosis by way of quenched reactive oxygen species (ROS) that may, otherwise, perpetuate NIS. Mechanistically, the selective and antagonistic effects of phytochemical agents and other dietary bioactive compounds on HNSCC in preclinical study designs have been previously described [37,38]. Moreover, results from a set of randomized control trials investigating the effects of *α*-tocopherol and *β*-carotene on radiation-induced toxicities in HNSCC patients suggested a potential therapeutic role of foods rich in those nutrients for mitigating NIS [39,40]. This hypothesis may be further substantiated by the results of analyses considering the low-carbohydrate index as the primary predictor, which was void of any significant association suggestive of a protective effect. Given that this index is composed of nutrient components rather than food groups, it is difficult to ascertain what types of foods contribute to higher adherence scores. Yet, it is valid to assume that the foods highest in fat and protein and lowest in dietary carbohydrates are those of animal origin, which include red and processed meats. Indeed, it was found in bivariate analyses (results not shown here) that total red and processed meat intake was positively correlated with the low-carbohydrate index (*r* = 0.28, *p* < 0.001) and that the index baring the strongest inverse relationship with red and processed meat intake was DASH (*r* = −0.38, *p* < 0.001). Red and processed meats are inherently devoid of phytochemical constituents that produce the anti-inflammatory effects we discuss and may, instead possess proinflammatory potential [40]. Interestingly, we did find a significant positive association between closer adherence to the low-carbohydrate index and increased odds of having dysphagia accompanying liquids, though there was no significant association with total symptom burden 1-year postdiagnosis. These results were similar to those reported between the “western” dietary pattern and the same outcome studied here in our analysis of a posteriori-derived dietary patterns [27].

There are some limitations to our study that are worth stating. Though the longitudinal design is a strength of the study, the use of baseline data to predict outcomes 1-year postdiagnosis may be confounded by diet changes implemented within that period. It is germane to posit that several participants may have adopted “healthier” diet changes in the intervening time window following their diagnosis. This would, potentially, explain why significant associations within the low-carbohydrate index were not ascertained, for instance. Though the 2007 Harvard FFQ, utilized for dietary collection in this study, has been validated for use in the general population, it has yet to be validated for the HNSCC population. Moreover, recall bias and other systematic biases accompanying the FFQ as the principal means of collecting dietary data should not be overlooked. Likewise, although the UM Head and Neck QOL Questionnaire has been validated in this patient population, the use of the NIS symptom summary score has not. Lastly, as is the case with any observational study design, the possibility of residual confounding and reverse causality cannot be ruled out.

## 5. Conclusions

Dietary patterns adhering to the guidelines forwarded by the aMED or DASH protocols were significantly associated with decreased odds of aggregate NIS symptom burden 1-year postdiagnosis. Other indices, AHEI-2010 and a low-carbohydrate index, showed attenuated, null, or in some cases positive associations. In summary, our findings suggest that promoting consumption of a diet abundant in fruits, vegetables, whole grains, low-fat dairy, legumes, nuts, while minimal in red and processed meat and sodium levels may ameliorate aggregate symptom burden in newly diagnosed HNSCC patients.

## Figures and Tables

**Figure 1 nutrients-13-03149-f001:**
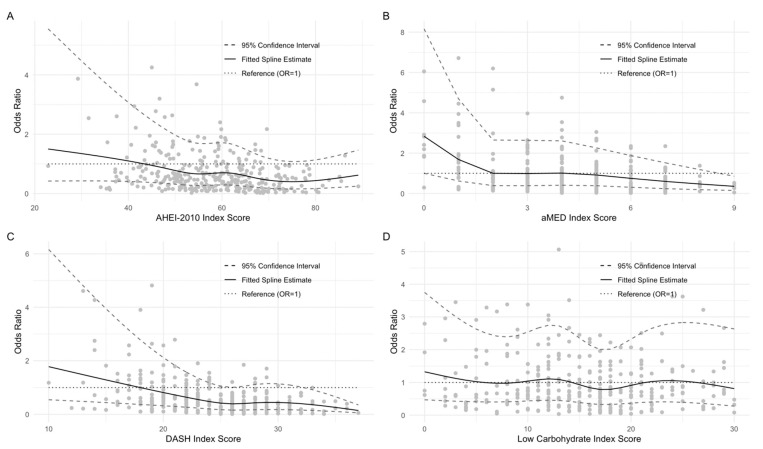
Dose–response relationship between each dietary index, modeled as a continuous variable, and NIS symptom summary score ≥12 1-year postdiagnosis. All multivariable models were adjusted for the same set of covariates in Table 5. Restricted cubic splines models were fit to mappings of each of the observations, according to their (**A**) AHEI-2010, (**B**) aMED, (**C**) DASH, or (**D**) low-carbohydrate index scores, to their respective odds. The odds corresponding to the median intake level for each dietary index were set as referents. Dashed lines indicate 95% confidence bounds. The dotted line indicates OR = 1, which was included for reference.

**Table 1 nutrients-13-03149-t001:** Demographic, clinical, and behavioral characteristics of the study participants (*n* = 323).

Characteristic		AHEI-2010		aMED		DASH		Low-Carbohydrate	
	Survivors # (%)	Q1 (*n* = 65)	Q5 (*n* = 64)	Q1 (*n* = 76)	Q5 (*n* = 46)	Q1 (*n* = 65)	Q5 (*n* = 59)	Q1 (*n* = 68)	Q5 (*n* = 57)
Age (y)									
Mean (SD)	60.4 (10.7)	56.6 (10.9)	62.1 (10.1)	58.1 (11.1)	59.8 (8.5)	57.9 (10.2)	62.3 (9.1)	57.2 (9.8)	62.3 (10.3)
Min/Max	29/95	29/78	34/83	29/85	34/83	30/78	43/81	30/78	43/85
Sex									
Male	254 (78.6)	52 (80.0)	46 (71.9)	65 (85.5)	37 (80.4)	57 (87.7)	45 (76.3)	51 (75.0)	40 (70.2)
Female	69 (21.4)	13 (20.0)	18 (28.1)	11 (14.5)	9 (19.6)	8 (12.3)	14 (23.7)	17 (25.0)	17 (29.8)
Education									
High school or less	91 (28.2)	25 (38.5)	11 (17.2)	24 (31.6)	5 (10.9)	31 (47.7)	7 (11.9)	20 (29.4)	16 (28.1)
Some college or more	232 (71.8)	40 (61.5)	53 (82.8)	52 (68.4)	41 (89.1)	34 (52.3)	52 (88.1)	48 (70.6)	41 (71.9)
Race/Ethnicity									
Non-Hispanic white	310 (96.9)	62 (95.4)	62 (96.9)	75 (98.7)	44 (97.8)	60 (93.8)	55 (94.8)	64 (94.1)	55 (98.2)
Other	7 (2.2)	3 (4.6)	1 (1.6)	1 (1.3)	1 (2.2)	4 (6.2)	2 (3.4)	2 (2.9)	1 (1.8)
Unknown	3 (0.9)	0 (0.0)	1 (1.6)	0 (0.0)	0 (0.0)	0 (0.0)	1 (1.7)	2 (2.9)	0 (0.0)
BMI (kg/m^2^)									
Underweight and normal weight (<25)	101 (31.3)	21 (32.3)	18 (28.1)	18 (23.7)	16 (34.8)	23 (35.4)	21 (35.6)	21 (30.9)	14 (24.6)
Overweight and obese (≥25)	222 (68.7)	44 (67.7)	46 (71.9)	58 (76.3)	30 (65.2)	42 (64.6)	38 (64.4)	47 (69.1)	43 (75.4)
Site									
Larynx	66 (20.4)	12 (18.5)	10 (15.6)	21 (27.6)	6 (13.0)	19 (29.2)	5 (8.5)	9 (13.2)	11 (19.3)
Oral cavity	96 (29.7)	22 (33.8)	19 (29.7)	23 (30.3)	14 (30.4)	16 (24.6)	13 (22)	20 (29.4)	22 (38.6)
Oropharynx	157 (48.6)	30 (46.2)	35 (54.7)	32 (42.1)	26 (56.5)	28 (43.1)	40 (67.8)	39 (57.4)	23 (40.4)
Hypopharynx	4 (1.2)	1 (1.5)	0 (0.0)	0 (0.0)	0 (0.0)	2 (3.1)	1 (1.7)	0 (0.0)	1 (1.8)
Stage									
0, I, II	104 (32.2)	18 (27.7)	20 (31.2)	26 (34.2)	13 (28.3)	20 (30.8)	14 (23.7)	24 (35.3)	18 (31.6)
III, IV	219 (67.8)	47 (72.3)	44 (68.8)	50 (65.8)	33 (71.7)	45 (69.2)	45 (76.3)	44 (64.7)	39 (68.4)
HPV Status									
HPV-negative	92 (28.5)	20 (30.8)	17 (26.6)	25 (32.9)	10 (21.7)	17 (26.2)	12 (20.3)	17 (25.0)	23 (40.4)
HPV-positive	71 (22.0)	8 (12.3)	15 (23.4)	16 (21.1)	13 (28.3)	11 (16.9)	15 (25.4)	20 (29.4)	7 (12.3)
Unknown	160 (49.5)	37 (56.9)	32 (50.0)	35 (46.1)	23 (50.0)	37 (56.9)	32 (54.2)	31 (45.6)	27 (47.4)
Treatment									
Surgery only	74 (22.9)	14 (21.5)	20 (31.2)	15 (19.7)	9 (19.6)	10 (15.4)	8 (13.6)	18 (26.5)	17 (29.8)
Surgery + adjuvant radiation	50 (15.5)	12 (18.5)	8 (12.5)	14 (18.4)	8 (17.4)	11 (16.9)	6 (10.2)	4 (5.9)	9 (15.8)
Radiation only	27 (8.4)	3 (4.6)	3 (4.7)	10 (13.2)	5 (10.9)	7 (10.8)	5 (8.5)	5 (7.4)	3 (5.3)
Chemotherapy + radiation	155 (48.0)	29 (44.6)	29 (45.3)	33 (43.4)	23 (50.0)	32 (49.2)	35 (59.3)	35 (51.5)	25 (43.9)
Chemotherapy only	7 (2.2)	2 (3.1)	2 (3.1)	3 (3.9)	1 (2.2)	2 (3.1)	2 (3.4)	2 (2.9)	2 (3.5)
Palliative or unknown	10 (3.1)	5 (7.7)	2 (3.1)	1 (1.3)	0 (0.0)	3 (4.6)	3 (5.1)	4 (5.9)	1 (1.8)
Smoking Status									
Current	106 (32.8)	34 (52.3)	11 (17.2)	33 (43.4)	8 (17.4)	37 (56.9)	11 (18.6)	24 (35.3)	16 (28.1)
Former	118 (36.5)	11 (16.9)	26 (40.6)	23 (30.3)	20 (43.5)	16 (24.6)	25 (42.4)	23 (33.8)	21 (36.8)
Never	99 (30.7)	20 (30.8)	27 (42.2)	20 (26.3)	18 (39.1)	12 (18.5)	23 (39)	21 (30.9)	20 (35.1)
Drinking Status									
Current	230 (71.2)	49 (75.4)	49 (76.6)	57 (75.0)	39 (84.8)	50 (76.9)	44 (74.6)	48 (70.6)	37 (64.9)
Former	71 (22.0)	11 (16.9)	13 (20.3)	15 (19.7)	6 (13.0)	14 (21.5)	13 (22)	18 (26.5)	16 (28.1)
Never	22 (6.8)	5 (7.7)	2 (3.1)	4 (5.3)	1 (2.2)	1 (1.5)	2 (3.4)	2 (2.9)	4 (7.0)

Percentages may not add to 100% given rounding.

**Table 2 nutrients-13-03149-t002:** Mean 1-year NIS summary score across select demographic, clinical, and behavioral characteristics (*n* = 323).

Characteristic	*n*	Mean NIS Summary Score (SD)	*p* ^a^
Age (y)			
≥60	169	13.6 (6.0)	0.09
<60	154	14.7 (6.2)	
Sex			
Male	254	13.9 (5.8)	0.18
Female	69	15 (7.2)	
Education			
High school or less	91	15.8 (6.4)	<0.01 **
Some college or more	232	13.5 (5.9)	
Race			
Non-Hispanic white	310	14.2 (6.2)	0.72
Other	7	13.9 (6.4)	
Unknown	3	13 (4.6)	
BMI (kg/m^2^)			
Underweight and normal weight (<25)	101	15 (6.6)	0.08
Overweight and obese (≥25)	222	13.7 (5.9)	
Site			
Larynx	66	12.4 (5.7)^†^	<0.01 **
Oral cavity	96	13.8 (6.5)	
Oropharynx	157	15.1 (6.0) ^†^	
Hypopharynx	4	13.2 (3.8)	
Stage			
0, I, II	104	11.9 (5.7)	<0.01 **
III, IV	219	15.2 (6.0)	
HPV Status			
HPV negative	92	14.4 (6.5)	0.44
HPV positive	71	14.5 (5.9)	
Unknown	160	13.8 (6.0)	
Treatment			
Surgery only	74	11 (5.2) ^†‡¥^	<0.0001 **
Surgery + adjuvant radiation	50	16.3 (6.7) ^†ψ^	
Radiation only	27	11.2 (4.5) ^ψξω^	
Chemotherapy + radiation	155	15.1 (5.7) ^‡ξ^	
Chemotherapy only	7	21 (8.0) ^¥ω^	
Palliative or unknown	10	15.3 (5.4)	
Smoking Status			
Current	106	15.3 (6.5)	0.58
Former	118	13.8 (6.0)	
Never	99	13.3 (5.7)	
Drinking Status			
Current	230	13.8 (5.9)	0.10
Former	71	15.3 (6.1)	
Never	22	13.7 (7.8)	

^a^*p*-value from ANOVA, modeling 1-year NIS summary score by indicated levels of characteristics; ** *p* < 0.01; In groups with >2 groups, means sharing a superscript (†, ‡, ψ, ξ, ω, or, ¥) in common are significantly different from one another.

**Table 3 nutrients-13-03149-t003:** Summary statistics for the diet quality index scores within this sample.

Index	Mean (SD)	Median	Minimum	Maximum	Theoretical (Max, Min)
AHEI-2010	57.99 (58)	58.54	22.85	89.14	(0, 110)
aMED	4.08 (58)	4	0	9	(0, 9)
DASH	23.98 (58)	24	10	37	(8, 40)
Low-Carbohydrate	14.98 (58)	15	0	30	(0, 30)

AHEI-2010: the Alternative Healthy Eating Index-2010; aMED: the Alternate Mediterranean Diet Index; DASH: the Dietary Approaches to Stop Hypertension.

**Table 4 nutrients-13-03149-t004:** Multivariable ^a^ ORs and 95% CI for association between quintiles of select diet quality index scores with being slightly to extremely bothered by at 1-year postdiagnosis (adjusted for baseline symptom levels) (*n* = 323).

Index/Symptom	Q1	Q2	Q3	Q4	Q5	*p* _trend_	*p* _Q5-Q1_
*AHEI-2010*	*n* = 65	*n* = 65	*n* = 64	*n* = 65	*n* = 64		
Trismus	1.00	0.90 (0.39–2.06)	0.93 (0.41–2.11)	0.51 (0.21–1.17)	0.65 (0.28–1.54)	0.16	0.33
Xerostomia	1.00	1.07 (0.37–3.04)	0.95 (0.32–2.76)	0.50 (0.17–1.38)	0.42 (0.14–1.21)	0.04 *	0.11
Difficulty chewing	1.00	0.93 (0.40–2.18)	0.86 (0.36–2.01)	0.38 (0.16–0.88) *	0.55 (0.24–1.26)	0.03 *	0.16
Dysphagia liquids	1.00	0.59 (0.26–1.32)	0.58 (0.26–1.28)	0.48 (0.21–1.09)	0.47 (0.19–1.09)	0.07	0.08
Dysphagia solids	1.00	0.77 (0.34–1.77)	0.77 (0.34–1.76)	0.44 (0.19–0.99) *	0.65 (0.28–1.50)	0.15	0.32
Taste	1.00	0.76 (0.31–1.87)	1.03 (0.41–2.63)	0.40 (0.16–0.96) *	0.49 (0.20–1.20)	0.04 *	0.12
Mucositis	1.00	1.45 (0.67–3.16)	1.19 (0.55–2.58)	0.64 (0.30–1.39)	0.81 (0.37–1.78)	0.18	0.61
NIS summary score ^b^	1.00	0.85 (0.37–1.93)	0.89 (0.39–2.03)	0.44 (0.20–0.99) *	0.65 (0.28–1.49)	0.12	0.31
*aMED*	*n* = 76	*n* = 55	*n* = 112	*n* = 34	*n* = 46		
Trismus	1.00	0.76 (0.33–1.72)	0.69 (0.34–1.39)	0.30 (0.09–0.85) *	0.84 (0.34–2.07)	0.27	0.71
Xerostomia	1.00	0.69 (0.26–1.85)	0.58 (0.24–1.39)	0.28 (0.08–0.95) *	0.30 (0.10–0.92) *	0.01 *	0.04 *
Difficulty chewing	1.00	0.71 (0.31–1.65)	0.58 (0.28–1.18)	0.29 (0.11–0.76) *	0.32 (0.13–0.79) *	<0.01 **	0.01 *
Dysphagia liquids	1.00	0.40 (0.18–0.88) *	0.37 (0.18–0.73) **	0.44 (0.17–1.13)	0.13 (0.04–0.38) **	<0.01 **	<0.01 **
Dysphagia solids	1.00	0.27 (0.11–0.61) **	0.40 (0.19–0.81) *	0.21 (0.07–0.56) **	0.34 (0.13–0.84) *	0.02 *	0.02 *
Taste	1.00	0.44 (0.18–1.07)	0.30 (0.14–0.65) **	0.57 (0.20–1.64)	0.64 (0.24–1.72)	0.60	0.37
Mucositis	1.00	0.69 (0.32–1.47)	0.82 (0.42–1.56)	1.23 (0.50–3.04)	0.39 (0.16–0.91) *	0.18	0.03 *
NIS summary score ^b^	1.00	0.36 (0.16–0.81) *	0.35 (0.17–0.72) **	0.33 (0.12–0.86) *	0.36 (0.14–0.88) *	0.04 *	0.03 *
*DASH*	*n* = 65	*n* = 83	*n* = 76	*n* = 40	*n* = 59		
Trismus	1.00	0.62 (0.28–1.36)	0.56 (0.25–1.24)	0.50 (0.18–1.33)	0.81 (0.34–1.93)	0.48	0.63
Xerostomia	1.00	0.34 (0.12–0.92) *	0.66 (0.23–1.89)	0.27 (0.08–0.86) *	0.27 (0.08–0.85) *	0.04 *	0.03 *
Difficulty chewing	1.00	0.48 (0.21–1.07)	0.40 (0.17–0.89) *	0.50 (0.18–1.33)	0.39 (0.16–0.95) *	0.06	0.04 *
Dysphagia liquids	1.00	0.48 (0.22–1.03)	0.51 (0.23–1.10)	0.49 (0.19–1.25)	0.37 (0.15–0.90) *	0.04 *	0.03 *
Dysphagia solids	1.00	0.46 (0.21–1.01)	0.43 (0.19–0.95) *	0.45 (0.17–1.18)	0.54 (0.22–1.28)	0.19	0.17
Taste	1.00	0.39 (0.15–0.93) *	0.29 (0.11–0.71) **	0.16 (0.05–0.44) **	0.50 (0.18–1.37)	0.04 *	0.18
Mucositis	1.00	0.84 (0.41–1.74)	0.54 (0.26–1.14)	0.61 (0.24–1.48)	0.61 (0.27–1.37)	0.12	0.23
NIS summary score ^b^	1.00	0.50 (0.22–1.11)	0.31 (0.13–0.68) **	0.35 (0.13–0.92) *	0.38 (0.15–0.91) *	0.02 *	0.03 *
*Low-Carbohydrate*	*n* = 68	*n* = 69	*n* = 72	*n* = 57	*n* = 57		
Trismus	1.00	1.13 (0.52–2.46)	1.12 (0.51–2.46)	0.60 (0.25–1.41)	0.68 (0.29–1.59)	0.20	0.37
Xerostomia	1.00	2.64 (0.94–7.68)	0.69 (0.26–1.81)	1.57 (0.54–4.69)	1.35 (0.47–3.93)	0.94	0.58
Difficulty chewing	1.00	0.87 (0.40–1.91)	1.17 (0.54–2.55)	0.43 (0.18–0.97) *	1.22 (0.53–2.84)	0.85	0.65
Dysphagia liquids	1.00	2.03 (0.89–4.72)	1.37 (0.60–3.19)	0.92 (0.37–2.26)	2.47 (1.06–5.91) *	0.19	0.04 *
Dysphagia solids	1.00	1.37 (0.63–2.98)	1.21 (0.56–2.65)	0.87 (0.39–1.96)	1.25 (0.55–2.83)	0.90	0.60
Taste	1.00	0.71 (0.30–1.66)	0.58 (0.25–1.33)	0.47 (0.19–1.16)	0.59 (0.24–1.42)	0.15	0.24
Mucositis	1.00	0.75 (0.36–1.57)	0.68 (0.33–1.40)	0.65 (0.30–1.39)	0.87 (0.40–1.89)	0.59	0.72
NIS summary score ^b^	1.00	1.66 (0.77–3.65)	1.24 (0.58–2.65)	0.80 (0.36–1.77)	1.25 (0.56–2.80)	0.92	0.59

^a^ Adjusted for age, tumor site, BMI, education status, cancer stage, smoking status, HPV status, BMI, total calories, and corresponding baseline symptom score; ^b^ Outcome modeled was NIS symptom summary score (generated by taking a subject’s sum of their individual NIS scores) ≥12 1-year postdiagnosis; * *p* < 0.05; ** *p* < 0.01; All evaluated NIS were measured using a discrete scale from “1” (indicating “not at all bothered”) to “5” (indicating “extremely bothered”) and dichotomized as “not at all bothered” and “slightly to extremely bothered”.

**Table 5 nutrients-13-03149-t005:** Multivariable ^a^ ORs and 95% CI for association between quintiles of select food groups and nutrient categories with NIS symptom summary score ≥12 1-year postdiagnosis (*n* = 323).

Food Group	Q1	Q2	Q3	Q4	Q5	*p* _trend_	*p* _Q5-Q1_
Legumes (servings/d)	1.00	0.87 (0.44–1.72)	0.79 (0.34–1.81)	0.83 (0.38–1.83)	0.69 (0.33–1.43)	0.34	0.32
Nuts (servings/d)	1.00	0.68 (0.31–1.46)	0.56 (0.25–1.23)	0.61 (0.28–1.32)	0.33 (0.15–0.72) **	0.01 *	0.01 **
Whole Grains (g/d)	1.00	0.80 (0.36–1.74)	0.73 (0.33–1.58)	0.61 (0.27–1.37)	0.61 (0.25–1.44)	0.24	0.26
Alcohol (g/d)	1.00	0.75 (0.34–1.62)	0.50 (0.23–1.06)	0.90 (0.42–1.92)	1.13 (0.52–2.48)	0.24	0.75
Red and Processed Meats (servings/d)	1.00	1.94 (0.91–4.18)	1.58 (0.75–3.32)	1.87 (0.83–4.27)	1.48 (0.63–3.47)	0.54	0.37
Total Fruit (servings/d)	1.00	0.53 (0.23–1.19)	0.27 (0.12–0.59) **	0.41 (0.18–0.91) *	0.32 (0.14–0.74) **	0.03 *	0.01 **
Total Vegetables (servings/d)	1.00	1.65 (0.76–3.67)	1.41 (0.66–3.02)	0.90 (0.42–1.95)	1.32 (0.59–2.98)	0.95	0.50
UFA/SFA Ratio	1.00	1.16 (0.53–2.55)	0.63 (0.29–1.35)	0.86 (0.39–1.87)	0.66 (0.30–1.44)	0.22	0.29
Sugar-Sweetened Beverages (servings/d)	1.00	1.33 (0.60–2.92)	0.95 (0.44–2.04)	0.92 (0.42–2.03)	0.54 (0.23–1.24)	0.05 *	0.15
Total Low-Fat Dairy (servings/d)	1.00	0.81 (0.37–1.78)	0.62 (0.29–1.31)	0.65 (0.29–1.43)	0.59 (0.27–1.28)	0.27	0.18
Total Sodium (mg/d)	1.00	0.79 (0.35–1.73)	0.99 (0.41–2.39)	1.11 (0.41–3.03)	0.65 (0.17–2.38)	0.67	0.51
n-3 Fatty Acids (g/d)	1.00	1.05 (0.48–2.29)	0.93 (0.43–2.01)	0.75 (0.35–1.59)	0.51 (0.23–1.09)	0.04 *	0.08
Trans Fat (% of total kcal)	1.00	0.46 (0.21–0.99) *	0.68 (0.31–1.47)	0.77 (0.35–1.68)	1.10 (0.49–2.48)	0.50	0.82
Total Carbohydrate (g/d)	1.00	1.18 (0.52–2.70)	1.24 (0.49–3.15)	1.45 (0.48–4.41)	0.96 (0.22–4.24)	0.98	0.96
Total Protein (g/d)	1.00	1.09 (0.47–2.50)	0.94 (0.38–2.31)	0.90 (0.32–2.57)	0.47 (0.13–1.74)	0.19	0.26
Total Fat (g/d)	1.00	0.96 (0.42–2.19)	1.21 (0.50–2.95)	1.03 (0.36–2.94)	0.72 (0.18–2.88)	0.70	0.65

^a^ Adjusted for age, tumor site, BMI, education status, cancer stage, smoking status, HPV status, total calories, and baseline NIS symptom summary score; * *p* < 0.05; ** *p* < 0.01.

## Data Availability

The data presented in this study are available on request from the corresponding author. The data are not publicly available due to privacy concerns.

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
