# Peer review of "Pretreatment Adherence to a Priori-Defined Dietary Patterns Is Associated with Decreased Nutrition Impact Symptom Burden in Head and Neck Cancer Survivors"

_nutrients, 2021, doi:10.3390/nu13093149_

Round 1

Reviewer 1 Report

The study presented is very interesting, well written and easy to read. The authors have evaluated how adherence to some of the dietary indices used in the literature are associated with the presence of some of the symptoms associated with head and neck cancers. 

Just some general comments that may clarify some aspects of the article. In the material and methods part it would be convenient the reason for selecting the alternative Mediterranean diet index or the alternative HEI and those four indexes instead of others like the IDM, HEI, Inflammatory Index etc....
In the results part I have not found Figure 1. 
Throughout the manuscript it would be advisable to moderate causality relationships. 

Author Response

The authors thank the reviewer for taking the time to evaluate our research article. Please see the attached uploaded revised manuscript. Edits to the manuscript can be seen as tracked changes. Please also note that the affiliation of one of our team members has been changed as well.

Reviewer comment: “Just some general comments that may clarify some aspects of the article. In the material and methods part it would be convenient the reason for selecting the alternative Mediterranean diet index or the alternative HEI and those four indexes instead of others like the IDM, HEI, Inflammatory Index etc....”

Authors’ response: Thank you for this suggestion. The methods section of the manuscript has been updated with a more nuanced explanation for the choice of the diet quality indices examined in lines 140-142. Moreover, we recognize the existence of other validated indices that may use proprietary algorithms and we a made a decision to evaluate indices that have been well studied and whose algorithms are public knowledge.

Reviewer comment: “In the results part I have not found Figure 1.”

Author’s response: We thank the reviewer for bringing this to our attention and for reminding us to double check our figures and references to those figures in the manuscript text. We invite the reviewer to see lines 359-361 in the revised manuscript, which is where we make reference to Figure 1. We also realize that the formatting of the text with the original figure image embedded may have been altered through the online submission process. We have revised this and re-embedded the figure once again into the text and should be visible on page 11. Please advise if you are still not able to see the figure image.

Reviewer comment: “Throughout the manuscript it would be advisable to moderate causality relationships.”

Authors’ response: We thank the reviewer for pointing this out to us. We have gone back and changed language in certain sections of the text to ensure that our results, in no way, imply causality. To our knowledge, throughout the manuscript we used key words such as “association” or “is associated with” to denote associations and not causal relationships. Please advise in a line-specific manner if you have any additional sections that would need to be changed to ensure causality is not implied. We also make mention of this in our limitations section (see lines 566-568).

Reviewer 2 Report

Thank you for the opportunity to review this manuscript on overall diet quality scores and nutrition impact symptoms (NIS) among patients diagnosed with head and neck squamous cell carcinoma (HNSCC). This is the first study to investigate this association using a priori defined diet quality index among this patient population. The study’s strengths include using a longitudinal cohort study to collect dietary and other information using standardized questionnaires before treatments started and to supplement information through medical records during the study follow-up period as well as investigating multiple overall diet quality scores and detailed stratified analyses. The authors report stronger associations of NIS with overall diet quality scores especially DASH and aMED than individual food groups nor nutrients. In general, the manuscript is well written. I have the following comments to strengthen and clarify the manuscript:

Methods - Statistical analysis:

The patients were recruited over six years (lines 88-90). It is possible that HNSCC treatment regimen as well as the popularity of diets have changed. The authors may need to consider calendar years in their analyses.

In lines 492-498, the authors highlight the importance of considering the amount of alcohol consumed, not only drinking status (never, former or current) as presented in Table 1. To compare directly with results reported by Potash et al (reference 37), please include the proportion of social drinkers and problem drinkers based on their definitions and conduct similar analyses and include the results in the revised manuscript.

Results:

In tables, six level of NIS were presented. It is helpful to include a footnote on which cutoff was used for six levels of NIS as described in lines 219-211.

For Table 3, it is helpful to have the possible range of the scores as they are not all can range from 0 to 100, perhaps as a footnote.

For Figure 1, it is possible that I may have missed some information, the median score does not seem to have OR of 1, although it was the reference. Please clarify or explain more in text.

Author Response

The authors thank the reviewer for taking the time to evaluate our research article. Please see the attached uploaded revised manuscript. Edits to the manuscript can be seen as tracked changes. Please also note that the affiliation of one of our team members has been changed as well.

Reviewer comment: “The patients were recruited over six years (lines 88-90). It is possible that HNSCC treatment regimen as well as the popularity of diets have changed. The authors may need to consider calendar years in their analyses.”

Authors’ response: We thank the reviewer for considering these key points and bringing them to our attention. We note that the longitudinal cohort study tracked treatment regimens of participants. As such, we included treatment regimen (which was recorded categorically) as a covariate that we adjusted for in our analyses to account for any influence that treatment regimen may have had on the outcome of NIS or in adherence to any of the analyzed diet indices. With regards to the reviewer’s second point, we believe that the change in the general public’s adherence to fad diets is very likely to not have much sway for newly diagnosed head and neck cancer survivors undergoing intensive treatment in the years proximal to their diagnosis. Moreover, given that this analysis measured adherence to a set of diet quality indices at study entry and was not concerned with analyzing changes in dietary intake throughout the early treatment course of head and neck cancer survivors, we feel that adding such an analysis would deter from the main analytical goal of this study.

Reviewer comment: “In lines 492-498, the authors highlight the importance of considering the amount of alcohol consumed, not only drinking status (never, former or current) as presented in Table 1. To compare directly with results reported by Potash et al (reference 37), please include the proportion of social drinkers and problem drinkers based on their definitions and conduct similar analyses and include the results in the revised manuscript.”

Authors’ response: We thank the author for bringing this detail to our attention. The SPORE longitudinal cohort study has implemented several ways of adjusting for drinking status and, in the years since its inception, modeling drinking status as “never”, “former”, or “current” has become the mainstay in analyses of these cohort data. Data were collected on problem drinking using the Alcohol Use Disorders Identification Test (AUDIT), but only a small proportion of the cohort reported problem drinking and we have found over the years that drinking status modeled as we did in this analysis is a more meaningful alcohol variable when used as a covariate. We also remind the reviewer that, unlike the Potash et al. reference, our analysis was not focused on drinking patterns in HNSCC as the primary variable of interest. We believe that shifting the analysis to feature analyses involving alcohol consumption would deter from the primary analytical goal of the study, which was to evaluate dietary patterns. In the manuscript we communicate that a decision was made to remove drinking status as a covariate since two of the indices have an alcohol component as part of their computational algorithms and because of the fact that smoking (which we include as a covariate) and drinking are highly correlated in the HNSCC population and in our sample (we did not want to overadjust and have issues with collinearity). Nevertheless, we have conducted analyses with the additional drinking covariate and the results do not change when we add alcohol consumption as a covariate into our models. We can provide these results upon request.

Reviewer comment: “In tables, six level of NIS were presented. It is helpful to include a footnote on which cutoff was used for six levels of NIS as described in lines 219-211.”

Authors’ response: We thank the reviewer for making this detail known to us. Table 4 has been updated accordingly with the reviewers’ recommendation.

Reviewer comment: “For Table 3, it is helpful to have the possible range of the scores as they are not all can range from 0 to 100, perhaps as a footnote.”

Authors’ Response: Thank you for this reminder. We have added a new column to table 3 that documents the theoretical maxima and minima for each respective index.

Reviewer comment: “For Figure 1, it is possible that I may have missed some information, the median score does not seem to have OR of 1, although it was the reference. Please clarify or explain more in text.”

Authors’ response: We thank the reviewer for allowing us to double check this matter. We invite the reviewer to see lines 278-280 in the revised uploaded manuscript where it reads: “The median of the lowest quintile of intake for each diet quality index was used as the referent value when computing odds ratios from these models.”
